# Antibiotic Augmentation of Thermal Eradication of *Staphylococcus epidermidis* Biofilm Infections

**DOI:** 10.3390/pathogens13040327

**Published:** 2024-04-16

**Authors:** Haydar A. S. Aljaafari, Nadia I. Abdulwahhab, Eric Nuxoll

**Affiliations:** 1Department of Chemical and Biochemical Engineering, University of Iowa, Iowa City, IA 52242, USA; haydar-aljaafari@uiowa.edu (H.A.S.A.); nadia.i.abdulwahhab@uotechnology.edu.iq (N.I.A.); 2Department of Chemical Engineering, University of Technology, Baghdad 10066, Iraq; 3Department of Applied Sciences, University of Technology, Baghdad 10066, Iraq

**Keywords:** biofilm, *Staphylococcus epidermidis*, hyperthermia, thermal shock, antibiotics, prothesis-related infections

## Abstract

*Staphylococcus epidermidis* is a major contributor to bacterial infections on medical implants, currently treated by surgical removal of the device and the surrounding infected tissue at considerable morbidity and expense. In situ hyperthermia is being investigated as a non-invasive means of mitigating these bacterial biofilm infections, but minimizing damage to the surrounding tissue requires augmenting the thermal shock with other approaches such as antibiotics and discerning the minimum shock required to eliminate the biofilm. *S. epidermidis* biofilms were systematically shocked at a variety of temperatures (50–80 °C) and durations (1–10 min) to characterize their thermal susceptibility and compare it to other common nosocomial pathogens such as *Staphylococcus aureus* and *Pseudomonas aeruginosa*. Biofilms were also exposed to three classes of antibiotics (ciprofloxacin, tobramycin and erythromycin) separately at concentrations ranging from 0 to 128 μg mL^−1^ to evaluate their impact on the efficacy of thermal shock and the subsequent potential regrowth of the biofilm. *S. epidermidis* biofilms were shown to be more thermally susceptible to hyperthermia than other common bacterial pathogens. All three antibiotics substantially decreased the duration and/or temperature needed to eliminate the biofilms, though this augmentation did not meet the criteria of synergism immediately following thermal shock. Subsequent reincubation, however, revealed strong synergism on a longer timescale.

## 1. Introduction

Infections associated with bacterial biofilms are a common health problem that is steadily increasing with the increased use of medical implants [1]. Each year, more than 1.1 million replacement hips and knees are surgically implanted in the United States [2], and this number is expected to increase exponentially over the next decade [3]. These implants may be colonized by bacteria forming a biofilm on the implant surface, enclosed within an extracellular polymeric matrix. Within the biofilm, bacteria are protected against antibiotics and the host’s immune system and have significantly different behaviors, making the treatment of these infections very challenging [4]. The current standard of care is the surgical removal of the infected implant and surrounding tissue, followed by replacement with a new device. [5]. While this may be performed in a single operation, the procedures are often separated by weeks of systemic antibiotic administration to clear bacteria from the rest of the body. Although this procedure is successful in most cases, replacement implants show a higher incidence rate of infection [6].

*Staphylococcus epidermidis* (*S. epidermidis*) has become the most important cause of nosocomial infections in recent years [7,8], so its susceptibility to any new mitigation approach is key to that approach’s success. *S. epidermidis* is normally commensal with human skin and mucous membranes, and typically, about 17 different strains inhabit healthy people at any one time [9]. This wide dermal prevalence likely contributes to its propensity for entering the body during device implantation procedures. Though *S. epidermidis* has been considered relatively innocuous, it is now accepted as a pathogenic bacterium [10], and its ability to adhere and form biofilm infections on foreign bodies is considered a virulence factor [7]. The physical harm, poor quality of life, and high economic burden of the current standard of care underscore the need for a new treatment approach to biofilm infections, particularly against *S. epidermidis*.

Remotely induced hyperthermia has shown promising results in eliminating bacterial biofilms [11,12,13,14,15,16,17,18,19]. An implant coating containing immobilized magnetic nanoparticles, for instance, can be heated by applying an alternating magnetic field from outside the body [20]. As any biofilm infection would be growing directly on the coating, this thermal shock can be applied to the biofilm precisely to mitigate it, potentially in combination with other approaches such as antibiotics, without surgical intervention. This thermal shock must propagate, however, into the surrounding tissue, which is more thermally susceptible than the biofilm itself. While hopefully less damaging than the current standard of care, this thermal shock must be kept to the minimum temperature and duration to eliminate the biofilm. Thus, characterizing the thermal susceptibility of these biofilms is a critical step in advancing this approach to mitigation. Studies have also suggested that antibiotics, while unable to eliminate the biofilm on their own, enhance the thermal susceptibility of some biofilm bacteria such as *Staphylococcus aureus* (*S. aureus*), reducing the shock needed to eliminate those biofilms and minimizing concurrent damage to adjacent tissue [17,21,22,23]. This paper reports the thermal susceptibility of *S. epidermidis* in conditions directly comparable to prior studies of *S. aureus* [23] and *Pseudomonas aeruginosa* (*P. aeruginosa*) [24] to determine its comparative susceptibility and investigates the effect of three different classes of antibiotics on this thermal susceptibility.

## 2. Materials and Methods

### 2.1. Streak and Inoculum

*S. epidermidis* coagulase-negative strain (University of Iowa Collections, #MNSCH) was streaked on an agar plate (Lennox LB Agar, Research Products International, Mt. Prospect, IL, USA) and incubated inverted for 24 h at 37 °C. A single isolated colony from the streaked plate was harvested using a sterile inoculating loop and moved into 5 mL (30 g L^−1^) Tryptic Soy Broth (TSB, Becton, Dickinson and Company, Franklin Lakes, NJ, USA). The inoculated TSB was incubated for 24 h at 37 °C to form *S. epidermidis* inoculum with an average of ~10^8.9^ colony-forming units (CFUs) per mL.

### 2.2. Culturing Biofilms

*S. epidermidis* biofilms were cultured using the Minimum Biofilm Eradication Concentration (MBEC) assay method (Innovotech, Edmonton, AB, Canada). This assay uses a 96-well base and a 96-peg lid, from which each peg is immersed into the center of a corresponding well in the base. A total of 1 mL of inoculum (~10^8.9^ CFU/mL) was diluted with 15 mL of TSB (30 g L^−1^), and 150 μL of this diluted inoculum was added to each well. The lid was sealed to the base with parafilm and horizontally incubated on an orbital shaker table (VWR 1000, 15 mm orbit, Radnor, PA, USA) set at 160 rpm at 37 °C for 24 h to culture MBEC biofilms.

### 2.3. Thermal Shock

MBEC biofilms were exposed to thermal shock at six different protocols (50 °C for 10 min; 60 °C for 1, 5 or 10 min; 70 °C for 1 or 5 min; and 80 °C for 1 min) by transferring the 96-peg lid holding the biofilms to a preheated 96-well base with 200 μL TSB (3 g L^−1^) in each well at the target temperature. This temperature was achieved and maintained by holding the base in a thermostatted water bath prior to and during thermal shock. Directly after the thermal shock, the biofilms were transferred to a new 96-well base with 200 μL TSB (3 g L^−1^) in each well at ambient temperature for enumeration.

### 2.4. Antibiotic Preparation

A total of 5 mg mL^−1^ ciprofloxacin stock was prepared by dissolving ciprofloxacin hydrochloride (MP Biomedicals, Santa Ana, CA, USA) in deionized water. Tobramycin stock was prepared following the same procedure by dissolving tobramycin sulfate salt (Sigma Aldrich, St. Louis, MO, USA). Erythromycin was obtained from MP Biomedicals, and the stock was prepared by mixing 1.5 mg mL^−1^ erythromycin into deionized water. Stocks were filtrated using a 0.22 μm pore PES membrane sterile filter (Millex^®^GP filter unit) and stored at 2 °C.

### 2.5. Antibiotic Exposure

To test planktonic *S. epidermidis*’s susceptibility to antibiotics, the inoculum (~10^8.9^ CFU/mL) was diluted at a ratio of 1:15 using TSB (30 g L^−1^), and then 150 μL of the diluted inoculum was added into each well of a 96-well base. After that, assays of desired antibiotic concentrations (0, 0.25, 1, 4, 16, 64 and 128 μg mL^−1^) were formed by adding the appropriate amount of antibiotics to each well, and dishes were sealed and incubated on a shaker table set at 160 rpm at 37 °C for 4 or 24 h.

Biofilms of *S. epidermidis* were rinsed for 1 min by placing the biofilms growing on the 96-peg lid into a 96-well base with 200 μL TSB (3 g L^−1^) in each well to remove unattached bacteria. After the rinse step, the biofilms were transferred along with their pegs to a new 96-well base containing 200 μL solutions of antibiotics (0, 0.25, 1, 4, 16, 64 and 128 μg mL^−1^) diluted in TSB (30 g L^−1^). They were incubated on a shaker table set at 160 rpm at 37 °C for 4 or 24 h.

### 2.6. Antibiotic and Thermal Exposure

To investigate the synergistic effect of antibiotics and thermal shock on the reduction in biofilm population density, MBEC biofilms were prepared and exposed to antibiotics (0.25, 4 or 128 μg mL^−1^) for four hours following the procedures described above. They were then transferred directly to a fresh 96-well base with the same array of antibiotic solutions, preheated to the targeted temperature and maintained as described above. After thermal shock at the desired temperature and exposure time (50 °C for 10 min; 60 °C for 1, 5 or 10 min; 70 °C for 1 or 5 min; and 80 °C for 1 min), biofilms were transferred to another fresh 96-well base with the same array of antibiotic solutions, and then they were incubated at 37 °C for the remainder of the 24 h total antibiotic exposure.

### 2.7. Enumeration

Biofilm population density was determined via resuspension and plating. Immediately following treatment, biofilms were separated from their surrounding planktonic bacteria by transfer into a new 96-well base with 200 μL TSB (3 g L^−1^) placed in each well. They were then disrupted and sonicated using a sonicator bath for 10 min at 45 kHz (VWR Symphony, 9.5 L). The sonicated homogenous solutions were then serially diluted in 10-fold increments and spot-plated in 10 μL samples on nutrient agar plates (Becton Dickinson and Company). After 10–15 min of spot plating and adsorption to the sample, plates were incubated inverted at 37 °C for 20–24 h. Grown colonies were counted, and the population density of colony-forming units (CFUs) per peg was found using Equation (1) [24], configured to express the population density on a logarithmic scale.
(1)$$\log\;(\frac{CFU}{peg})=\log\;[(\mathrm{Plate}\;\mathrm{count}\;\times \;10^{\mathrm{Dilution}\;\mathrm{factor}}\;\times \;(\frac{0.2\;mL}{0.01\;mL}/\mathrm{peg})\;+\;1]$$
where the following are used:-**Plate count**, which is the number of colonies observed in the plated sample.-**Dilution factor**, which is the number of tenfold dilutions needed to make that sample.-(**0.2 mL/0.01 mL**), which is the ratio of total biofilm suspension to sampled volume.-**(+1)**, which is added to ensure that the log ($\frac{CFU}{peg}$) value for samples with no CFUs is zero rather than mathematically undefined.

### 2.8. Statistical Analysis

The enumeration results were calculated on a log scale for statistical analysis. The key result discussed in this paper is the difference between the population decrease of a combination treatment and the product of the population decreases of the corresponding individual treatments. The significance of this was determined using two-tailed Student’s *t*-tests with a 95% confidence interval. Variance for each trial arm was assumed to be uncorrelated, and differences in variance were reconciled according to Cochran. Pairwise comparisons using ANOVA with Tukey/Kramer among all biofilms receiving only thermal shock (and their controls), among all planktonic samples treated only with an antibiotic (for each antibiotic), among all biofilms treated only with an antibiotic (for each antibiotic), and for each combination therapy vs. the corresponding individual therapies and untreated control have been tabulated at both the *p* < 0.05 and *p* < 0.01 levels and are available in the Appendix A, along with the numerical results of all trials.

## 3. Results

### 3.1. Biofilm Population and Thermal Susceptibility

*S. epidermidis* developed biofilms with an average population density of a million CFUs/peg (log (CFU/peg) = 10^6.00±0.52^) during 24 h of growth on MBEC pegs (negative control), as shown in Figure 1. The biofilms were already in the resting phase, with no statistically significant increase in population over an additional 20 h of reincubation. All thermal shocks except for the one at 50 °C for 10 min prompted a significant immediate decrease in population density, more pronounced with increasing temperature and duration. Upon 20 h of reincubation, however, all the biofilms subjected to 60 °C shocks recovered their resting phase population. No bacteria could be detected in 50% (12 of 24) or 87.5% (21 of 24) of the samples thermally shocked for 1 min at 70 °C or 80 °C, respectively. It seems that the biofilms were completely destroyed in these samples, as comparable percentages of biofilms subjected to those thermal shocks did not regrow after subsequent reincubation for 20 h, while the average population density of the surviving biofilms increased by two orders of magnitude. The biofilms that were thermally shocked at 70 °C for 5 min were uniformly destroyed.

### 3.2. Antibiotic Exposure without Thermal Shock

The three antibiotics chosen for this study vary widely in their efficacy against planktonic *S. epidermidis*, as shown in Figure 2. Panel A shows that, while the immediate (4 h) effect of ciprofloxacin on planktonic *S. epidermidis* is slight even at high (128 μg mL^−1^) concentrations, 20 h exposure decreases the population significantly even at therapeutically relevant (1 μg mL^−1^) concentrations, eliminating the bacteria entirely at grossly high concentrations. Tobramycin (Figure 2C) is more effective, eliminating the planktonic bacteria in only 4 h at 128 μg mL^−1^, though at clinically relevant concentrations (<16 μg mL^−1^), the bacteria regrow over the subsequent 20 h exposure. Erythromycin (Figure 2E) has only a modest impact at any concentration, with only marginal improvement over an additional day of exposure.

Against the biofilms, however, none of the antibiotics reduce the population more than a few orders of magnitude, even at grossly high concentrations. Moreover, for tobramycin (Panel D) and erythromycin (Panel F), the initial decrease seen after 4 h of exposure diminishes with time, suggesting that the biofilm will continue regrowing despite the presence of the antibiotic. This is not observed with ciprofloxacin (Panel B), though only a 2-log decrease is observed even at 128 μg mL^−1^.

### 3.3. Thermal Susceptibility with Antibiotics

Except for the mildest thermal shocks, exposure to antibiotics drove some biofilm populations below the detection limit upon thermal shock. Figure 3 shows the viability percentage for biofilms exposed to ciprofloxacin and thermally shocked, as well as the average population density for the surviving biofilms. Panel A confirms ciprofloxacin does not have any protective effect, with no biofilms surviving a 70 °C shock for 5 min, regardless of antibiotic concentration. Even under the mildest shock conditions (60 C for 1 min, Panel B), biofilm population density decreases with reincubation, which was not observed for biofilms without thermal shock or biofilms without ciprofloxacin exposure. Combined with ciprofloxacin at clinically relevant concentrations (4 μg mL^−1^), even moderate shocks (70 C for 1 min, 60 C for 10 min) eliminated all biofilms after 24 h.

The results for combined thermal shock and tobramycin exposure (Figure 4) are similar, despite tobramycin having a different mode of action. Tobramycin without heat shock did not prevent regrowth at any concentration, but tobramycin with any heat shock (even at 50 °C for 10 min, which had zero effect on its own) did prevent regrowth at 128 μg mL^−1^ and decreased viability. Lower doses of tobramycin (0.25 μg mL^−1^) were only effective with higher-temperature (80 °C) thermal shocks. Milder thermal shocks at 70 °C for 1 min or 60 °C for 10 min required higher doses of tobramycin (4 and 128 μg mL^−1^, respectively) for complete elimination.

Like tobramycin, erythromycin without heat shock did not prevent regrowth at any concentration except for the highest one (128 μg mL^−1^), but with heat shock, any dosage of erythromycin inhibited regrowth after any heat shock (Figure 5), even heat shocks that had no effect on their own (50 °C for 10 min). A thermal shock of 60 °C for 10 min was sufficient to eliminate all biofilms at clinical dosing (4 μg mL^−1^) within 20 h of thermal shock, though either treatment by itself resulted in only a 1-log reduction within 20 h of thermal shock. Thermal shocks at higher temperatures of 70 °C or 80 °C for 1 min required only 0.25 μg mL^−1^ erythromycin for complete elimination of the *S. epidermidis* biofilm.

## 4. Discussion

The use of heat to kill bacteria is well established but is generally used in great excess (e.g., autoclaving at 121 °C for 90 min) to ensure complete elimination. Such treatment is not feasible in vivo where tissue damage occurs above 43 °C [25]. Determining the least aggressive thermal shock necessary for eliminating an infection is critical to replacing surgical explantation and replacement as the standard of care for implant infections. While Figure 1 shows that these *S. epidermidis* biofilms are uniformly eliminated by a 70 °C shock for 5 min, combination with antibiotics shows elimination in only 1 min. Optimism for the feasibility of this approach is supported by reports of in vivo hyperthermia studies; for instance, a 75 °C shock for over a minute in a rat model did not show significant damage to the surrounding tissue [15]. 

The thermal susceptibility of a biofilm is not universally defined. One potential parameter is the decimal reduction time (D-time) for the thermal shock, defined as the duration of treatment required to reduce the population density by one order of magnitude. Dividing the exposure time by the decrease in log(CFU/peg) between the control and the thermally shocked biofilms in Figure 2, we see that the D-time is roughly 20 min at 70 °C, 2 min at 60 °C and 0.2 min at 50 °C, though comparisons of results at 60 °C for 1, 5 and 10 min show that the population does not maintain a strictly exponential decay. Thermal susceptibility also depends on many factors, such as the method by which the biofilm is cultured [14] and the mode by which the thermal shock is applied, making it challenging to compare results across studies. In this case, however, the culture method and mode of heat delivery have been designed to match previous studies with *P. aeruginosa* [24] and methicillin-resistant *S. aureus* [23], two other common nosocomial biofilm pathogens. 

As shown in Figure 6, *S. epidermidis* is far less thermally robust than *S. aureus*; a thermal shock designed to eliminate an *S. aureus* biofilm would ensure the elimination of *S. epidermidis* as well. Similarly, comparisons with *P. aeruginosa* results from the literature show that, while *P. aeruginosa* is unaffected by 70 °C thermal shocks for one minute, comparably grown and shocked *S. epidermidis* will decrease by two orders of magnitude, in some cases becoming nonviable (Figure 7). From this, we may conclude that *S. epidermidis* is more thermally susceptible than both *S. aureus* and *P. aeruginosa*, two other common pathogens in implant infections.

Previous studies have also shown that by achieving a critical population decrease in *P. aeruginosa*, the remaining biofilm would die off even after returning to incubation conditions [14,22], perhaps as a result of apoptotic activity poisoning the biofilm. In *S. epidermidis*, however, the percentage of biofilms with detectable CFUs does not decrease during a 20 h reincubation, indicating that this phenomenon is not present in *S. epidermidis*.

Antibiotics have long been the first line of defense against bacterial infections, but they cannot eliminate bacterial biofilms, as demonstrated in Figure 2. Three different classes of antibiotics were investigated, chosen in part for their thermal stability: ciprofloxacin (a fluoroquinolone), tobramycin (an aminoglycoside), and erythromycin (a macrolide). Ciprofloxacin seemed to eliminate planktonic *S. epidermidis*, albeit at grossly high concentrations and only after 24 h of exposure, while grossly high concentrations of tobramycin eliminated the planktonic bacteria much more quickly. Erythromycin showed little effect on *S. epidermidis* at any concentration. None of them could reduce the biofilm populations by more than a few orders of magnitude at any concentration, however. Moreover, the biofilms in tobramycin appeared to increase in population between hours 4 and 24 of exposure, though this effect was not significant at most concentrations.

Previous studies have reported synergism between heat and antibiotics in *P. aeruginosa, S. aureus* and *S. epidermidis* biofilms with different types of antibiotics [16,17,21,24], though the definition of ‘synergism’ is not universal. A simplistic but popular definition in this context requires only a population decrease caused by a combination treatment to be a certain amount (e.g., two orders of magnitude) greater than the population decrease caused by each individual treatment component. While simple to test for and quantify, such a definition is not useful for discerning mechanistic overlap between the treatments. For that, synergism requires an effect caused by a combination treatment that is greater than the product of the individual treatments. For instance, if one treatment reduces a biofilm by two orders of magnitude and another treatment reduces a biofilm by three orders of magnitude, then a biofilm population subjected to both treatments should decrease by five orders of magnitude even if the treatments operate completely independently, i.e., are orthogonal. A reduction of four orders of magnitude, while considered synergistic by the definition above, would actually indicate the opposite, i.e., interference between the two treatments. For instance, both treatments may inhibit the same protein as part of their mechanism of action. A synergistic treatment would drop the population by more than five orders of magnitude, indicating complementary mechanisms such as thermal shock inhibiting an efflux pump, which would otherwise protect bacteria from an antibiotic. While much more useful, observing synergism by this definition is also much more challenging, as the product of the effects of both treatments must not be greater than the difference between the initial biofilm population density (for this study, 10^6^ CFUs/peg) and the detection limit (for this study, 10^1.3^ CFUs/peg, defined by detecting a single colony in an undiluted sample in Equation (1)).

The thermal shocks for this study were chosen not only to demonstrate combination treatments that could eliminate these biofilms but also to provide population decreases that, when added to those of the antibiotics, would still result in quantifiable biofilms to investigate any mechanistic overlap between the two treatments. Multiple time points were used to account for the fact that some processes may occur on a different time scale than others. Ciprofloxacin at 128 μg mL^−1^, for example, decreases the biofilm population density in four hours by two orders of magnitude, and a 1 min thermal shock at 60 °C also decreases the biofilm by two orders of magnitude. Combined, however, they reduce the biofilm by only three orders of magnitude, which is less than if the two treatments were completely orthogonal. However, 20 h later, surviving thermally shocked biofilms will grow back, showing no decrease in population, while non-shocked biofilms in 128 μg mL^−1^ ciprofloxacin will decrease another order of magnitude in those additional hours. The combined treatments, if orthogonal, should result in a biofilm three orders of magnitude smaller than an untreated one. Figure 3 shows, however, that the biofilm is four orders of magnitude smaller. Combinations that appear antagonistic when measured immediately after the thermal shock are in fact synergistic over a longer time period. With ciprofloxacin, this effect is small, and there are only a few combinations where this antagonism (i.e., the experimentally observed population decrease being smaller than the decrease predicted by adding the population decreases of the individual treatments) can be stated with 95% confidence immediately after the thermal shock. Population decreases after 20 h, however, are significantly larger than the decreases predicted for the individual treatments after 20 h in almost all combinations where enough biofilms survived for quantification. At 20 h, thermal shocks and antibiotics are synergistic. The results for tobramycin are similar, with a few combinations suggesting antagonism immediately post-shock, but most with no significant distinction, while at 20 h post-shock, the populations are consistently significantly lower than those calculated by applying the decreases of the individual treatments. In erythromycin, the difference is more pronounced. Few biofilms are completely eliminated immediately after any erythromycin/thermal shock combinations, and their surviving population is significantly higher than that predicted by orthogonal effects in almost every combination. After 20 h, however, many more biofilms are eliminated, and the surviving ones have populations that are uniformly far smaller than those that would be predicted by the individual treatments sequentially. Immediately post-shock, erythromycin is antagonistic to thermal shock, while at a 1-day time scale, it is strongly synergistic. Interestingly, these effects were as likely to be seen with 0.25 μg mL^−1^ of antibiotic as with 128 μg mL^−1^. 

These results should not convince anyone that a clinical *S. epidermidis* infection on a medical implant will be reliably eliminated by a 1 min shock at 70 °C with 4 μg mL^−1^ of tobramycin but not by a 5 min shock at 60 °C with 0.25 μg mL^−1^ of ciprofloxacin. There are significant differences between these in vitro studies and clinical treatment, notably the presence of an immune system, which may contribute significantly to defeating the biofilm. On the other hand, these biofilms were thermally shocked by immersion in preheated media to provide instant, precise, highly controlled shocks, while clinical implementation would require heating the biofilm’s substrate. This would provide the bacteria with the opportunity to reversibly disperse into the (relatively) cooler tissue above, avoiding part of the thermal shock. One may conclude, however, that *S. epidermidis* biofilms may require a less aggressive thermal shock than biofilms of some other common pathogens and that, while antibiotics may not immediately augment the efficacy of thermal shocks for destroying biofilms, they do, in fact, enhance thermal shocks synergistically to eventually destroy the biofilm.

## 5. Conclusions

Bacterial biofilm infections are a major problem for medical implants, with the poor current standard of care involving multiple invasive surgeries. In situ hyperthermia is a new approach being investigated to provide a much safer, less invasive means of eradicating biofilm infections on implants. Thermally destroying biofilms in situ will also cause thermal damage to surrounding tissue, so knowing the minimum degree of thermal shock to apply is critically important. *Staphylococcus epidermidis* is one of the most common pathogens in these bacterial biofilm infections. This work has shown that *S. epidermidis* biofilms are more susceptible to thermal shock than *Staphylococcus aureus* or even *Pseudomonas aeruginosa* and can be eliminated with milder thermal shocks. Moreover, the application of antibiotics at clinically relevant doses can substantially further reduce the degree of thermal shock needed, regardless of the class of antibiotic used. While antibiotics may not appear synergistic immediately following thermal shock in *S. epidermidis*, they do act synergistically to mitigate biofilms.

## Figures and Tables

**Figure 1 pathogens-13-00327-f001:**
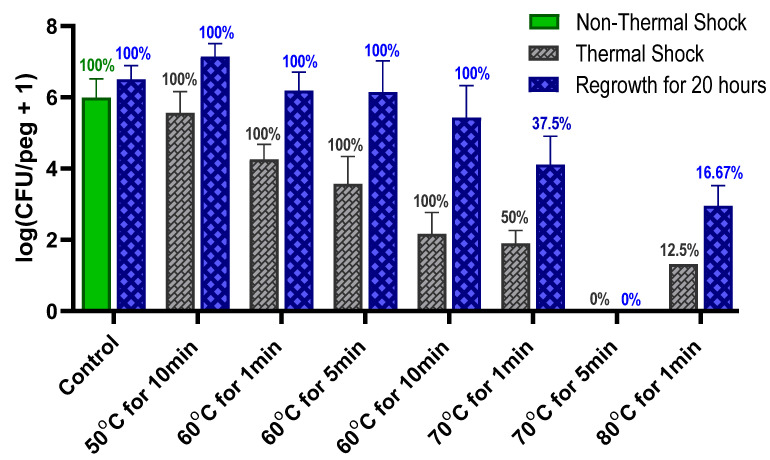
Effect of thermal shock and reincubation on *S. epidermidis* biofilms. Error bars indicate standard deviation for at least 6 pegs. Numbers above each bar indicate the percentage of pegs with detectable bacteria after the corresponding procedure.

**Figure 2 pathogens-13-00327-f002:**
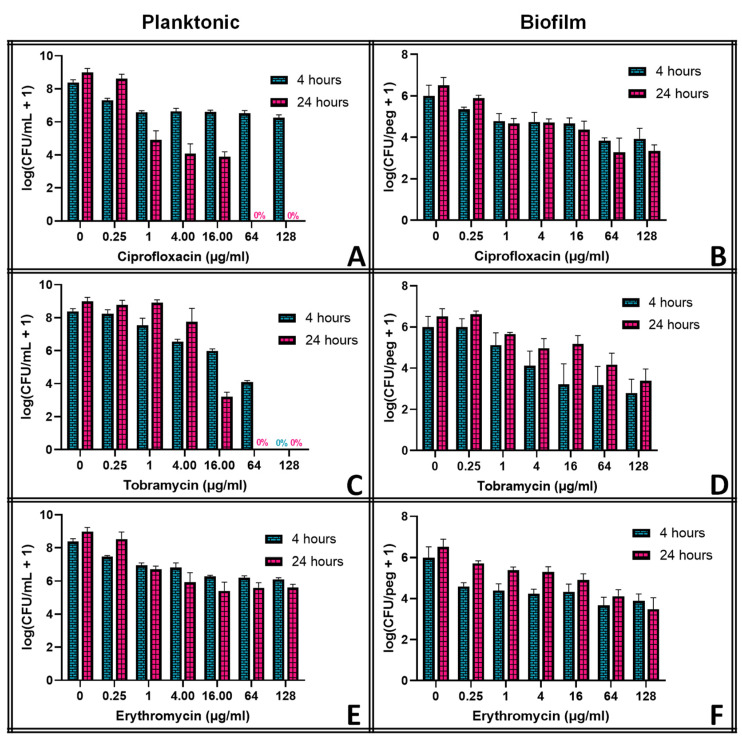
Effect of antibiotics on *S. epidermidis* in planktonic (panels (**A**,**C**,**E**)) and biofilm (panels (**B**,**D**,**F**)) phenotypes. Error bars indicate standard deviation for at least 4 pegs.

**Figure 3 pathogens-13-00327-f003:**
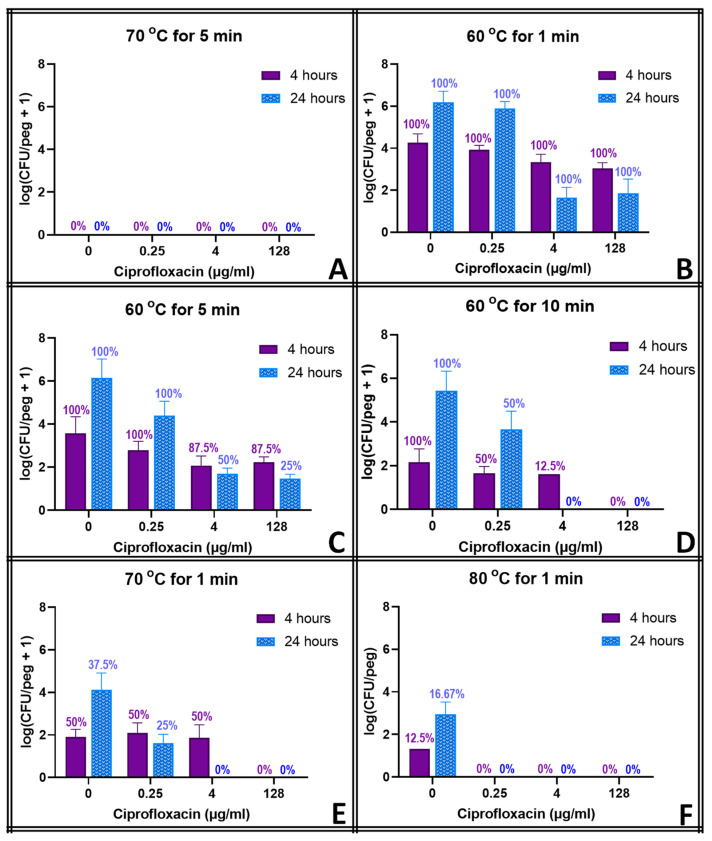
Combined effect of thermal shock and ciprofloxacin on *S. epidermidis* biofilms. Error bars indicate standard deviation for at least 6 pegs. Numbers above each bar indicate the percentage of pegs with detectable bacteria after the corresponding procedure. Each panel (**A**–**F**) represents a different temperature/exposure time combination, as indicated.

**Figure 4 pathogens-13-00327-f004:**
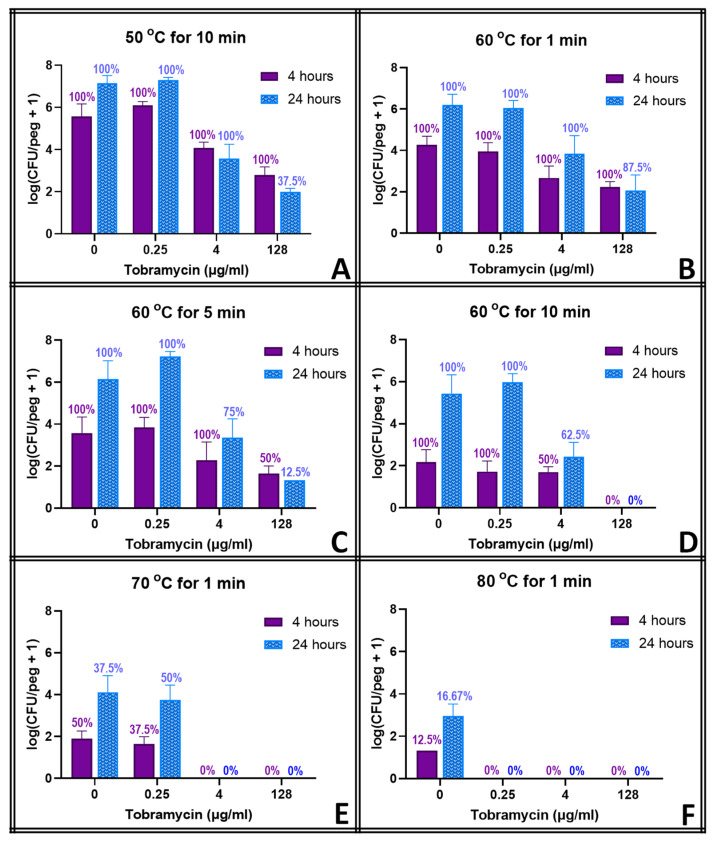
Combined effect of thermal shock and tobramycin on *S. epidermidis* biofilms. Error bars indicate standard deviation for at least 6 pegs. Numbers above each bar indicate the percentage of pegs with detectable bacteria after the corresponding procedure. Each panel (**A**–**F**) represents a different temperature/exposure time combination, as indicated.

**Figure 5 pathogens-13-00327-f005:**
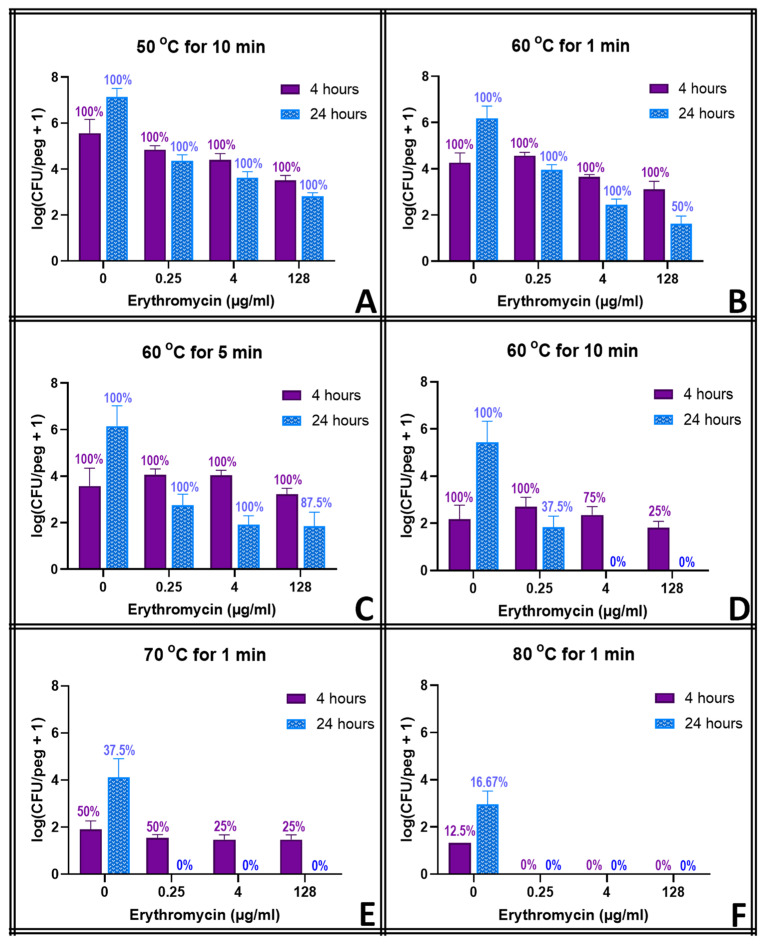
Combined effect of thermal shock and erythromycin on *S. epidermidis* biofilms. Error bars indicate standard deviation for at least 6 pegs. Numbers above each bar indicate the percentage of pegs with detectable bacteria after the corresponding procedure. Each panel (**A**–**F**) represents a different temperature/exposure time combination, as indicated.

**Figure 6 pathogens-13-00327-f006:**
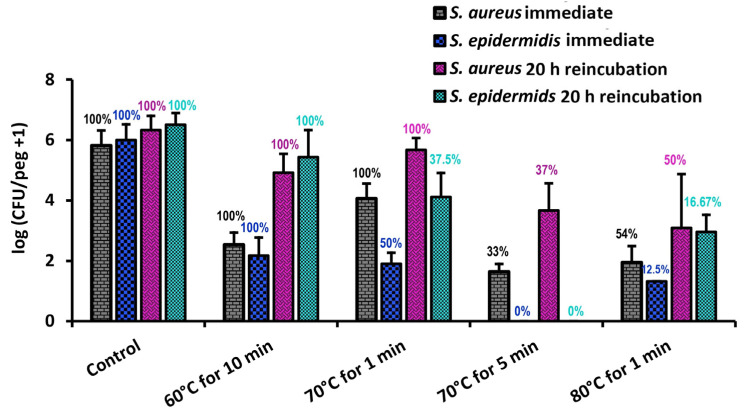
Comparison of *S. aureus* and *S. epidermidis* biofilm thermal susceptibilities. Error bars indicate standard deviation for at least 6 pegs. Numbers above each bar indicate the percentage of pegs with detectable bacteria after the corresponding procedure. *S. aureus* results have been taken from ref. [23].

**Figure 7 pathogens-13-00327-f007:**
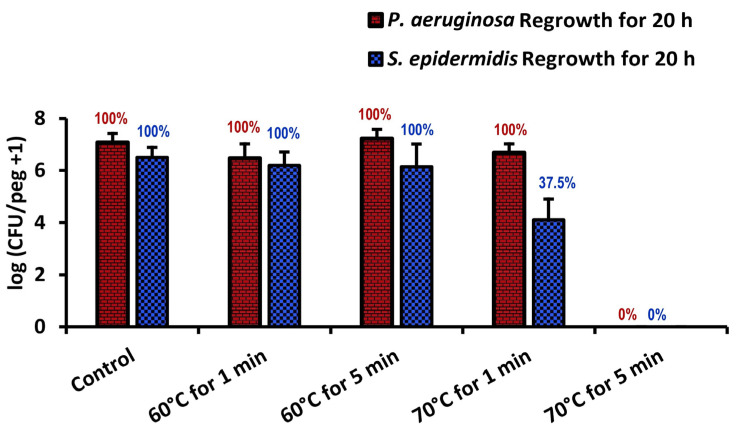
Comparison of *P. aeruginosa* and *S. epidermidis* biofilm thermal susceptibilities. Error bars indicate standard deviation for at least 6 pegs. Numbers above each bar indicate the percentage of pegs with detectable bacteria after the corresponding procedure. *P. aeruginosa* results have been taken from ref. [24].

## Data Availability

All data generated in this study are available in the Appendix A for this manuscript.

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
