# Peer review of "Antibiotic Augmentation of Thermal Eradication of Staphylococcus epidermidis Biofilm Infections"

_pathogens, 2024, doi:10.3390/pathogens13040327_

Round 1

Reviewer 1 Report

Comments and Suggestions for Authors

The manuscript presented for review is well-written and structured report of the antibiotic augmentation of thermal eradication of Staphylococcus epidermidis biofilm infections. Staphylococcus epidermidis strains pose a serious treatment problem among hospitalized patients. Moreover, biofilm infections are now becoming more common. Therefore, the subject of presented manuscript raises an especially important clinical problem.

The manuscript is interesting, complete and well structured, but it is necessary to do major revision to be accepted:

Major specific suggestion/comments:

Line 53: Are there any newer reports in this area?

Line 73: MNSCH? Shouldn't there be a CNS? the abbreviation is also not used in the rest of the text so it can be removed.

Line 98: deionized

Figure 6 and 7: species names should be in italics.

Line 25: italic

Line 42: Staphylococcus (S.) epidermidis à Staphylococcus epidermidis (S. epidermidis)

Line 62 and 68: as above

Line 340: highly-controlled à highly controlled

Please use abbreviations consistently in whole text.

Line 158: 87.5 % à 87.5%

Line 159: 70 °C à 70°C - please correct throughout

Line 251: P .aeruginosa à P. aeruginosa

Reviewer 2 Report

Comments and Suggestions for Authors

The study conducted by Aljaafari et al., assessed the impact of antibiotics solely and in combination with different levels of thermal shock on the elimination of Staphylococcus epidermidis biofilms. The study may have practical implications for future design of non-invasive approach for eliminating S. epidermidis biofilm. However, there are several concerns to be addressed,

1.       Major:

·         Section 2.6: Why did authors not consider sequential application of thermal shock, then antibiotic exposure?

·         Section:  2.7: for enumeration, instead of spot tittering which provides estimates of the populations, why a standard plate counting has not been performed.

·         Section 2.7: The rational for using “+1” is not understood, please justify.

·         The statistical analyses need to consider comparing all groups and performing pairwise comparisons. ANOVA with Tukey test may suffice.

·         All figures need to show statistical analysis output to show differences among the groups.

·         Transform data in the results section to Log10 CFU/peg, instead of CFU/peg.

·         Figure 2:  What is the methodology used for differentiating population counts of planktonic versus biofilms cells?

2.       Minor

·         Line 81: what does MBEC stand for?

·         Line 174: “eliminating the biofilm…”, do you mean planktonic?

·         Rephrase some statements for clarity. Line 189-190; Line 209-211; Line 243-252 (Revise bacterial nomenclature, e.g., Pseudomonas aeruginosa should be abbreviated, and add space between “S” and “aureus”; “S” and “epidermidis” and so on.

·         Revise bacterial nomenclature on Figure 6 and 7 (italicization, spaces between genus and species names).

·         Rephrase Line 259-262, and Line 275-277.

·         Conclusion: Emphasize How antibiotic-thermal shock combination can be verified in in vivo models before further applications in clinical settings.

Comments on the Quality of English Language

Some statements need to be rephrased in the results and discussion section for clarity. These statements have been specified in my review report. 

Reviewer 3 Report

Comments and Suggestions for Authors

Reviewer Comments

·       Key words: Staphylococcus epidermidis / Staphylococcus epidermidis (Italicize)

·       Introduction – Lines 63-64: Recent has also suggested that antibiotics, while unable to eliminate the biofilm on its own, … / Recent studies have also suggested that antibiotics, while unable to eliminate the biofilm on its own, …

·       Materials and methods - Lines 78 and 83: …108.9 Colony Forming Units (CFU) … / How is the CFU a decimal?

·       Materials and methods – Line 81: S. epidermidis biofilms were cultured using MBEC assays … / How do you culture Staph using MBEC assay? Please rephrase either as S. epidermidis biofilms were cultured using MBEC method or S. epidermidis biofilms were cultured as per the MBEC assay method. Also please expand MBEC.

·       Materials and methods - Line 89: MBEC biofilms were thermally shocked at six different protocols … / MBEC biofilms were exposed to thermal shock at six different protocols.

·       Materials and methods - Line 102: … were filtrated using 0.22 μL PES membrane … / Please correct as 0.22 µm as it is the pore size of the membrane.

·       Results - Line 163: Biofilms thermally shocked at 70 C for 5 minutes … / 70oC

·       Results – Lines 189-190: Except with the mildest thermal shocks, exposure to antibiotics caused drove some biofilm populations below the detection limit upon thermal shock. / Please rephrase this sentence.

·       Results – Line 210: … °C for 10 min required higher doses of tobramycin for complete elimination 4 or 128 μg ml-1, respectively. … / … °C for 10 min required higher doses of tobramycin for complete elimination either 4 or 128 μg ml-1, respectively.

·       Discussion – Line 229-230: The use of heat to kill bacteria is well-established, but is generally used in great excess (e.g. autoclaving at 121 °C for 90 min) to ensure complete elimination.

Conventional sterilization is carried out at 121oC for 15 minutes for liquid media however, for solid material you could go up to 60 minutes or opt for tyndallisation.

·      It would have been nice if the authors could have arrived at the Decimal Reduction Time (D value) of S. epidermidis at different temperatures with and without the antibiotics.

Reviewer 4 Report

Comments and Suggestions for Authors

The reviewed paper by Haydar A.S. Aljaafari et al tells about Staphylococcus epidermidis anti -biofilm effects of antibiotics coupled with remotely-induced hyperthermia. The idea is publication-worth, the paper is well written, experiments are well done and data are well discussed. Authors made an excellent work and used various approaches to characterize the effects. All obtained data are relevant and clearly show the antimicrobial potential of the enzyme. Nevertheless, some issues should be addressed before considering for publication

Why erythromycin has been used? This antimicrobial is out of the use to date. As a solution, authors should ad a phrase that they checked 3 different classes of antibiotics - fluorquinolones, aminoglycosides and macrolides with a one model antimicrobial within them. 

Major

This is not correct to compare the efficiency of biofilm elimination for  S. epidermidis to prior studies of S. aureus [20] and Pseudomonas aeruginosa [21], Only data obtained in one experiment can be compared, since conditions will be apriory different. Therefore authors should smooth their postulates and figs 6 and 7 are overestimation. 

Authors should add information about bacterial suscebtibility to antibiotics used, i.e. MIC and MBC values. 

The efficiency of treatment generally is belived to be sucsessfull if 3-log reduction of CFUs is achieved. This is recommended to re-analyze data taking into account this threshold. AS well, when comparing statistical significance between CFUs count in such assays, significant difference is observed at 2.7-log reduction of CFUs when using Pearson Chi squared homogenejty test. 

Thus, Fig 1 A - Cip has antimicrobial effect at 1 ug/ml for 24 h treatment of  planctonic cells and  at 64 ug/ml for 24 h treatment of biofilm-embedded cells. No significant effect can be achieved at 4 h treatment at concentrations used. This is recommended to show by asterisks those points where significant difference with untreated sample is observed 

Minor

References 1-15 are outdated. This is recommended to add more recent references to introduction section (2020-2024 years). Thus, "Recently, remotely-induced hyperthermia has shown promising results in eliminat- 53

ing bacterial biofilms [10-15]" - these are data published up to 2017, 7 years ago. 

line 29 Infection by bacterial biofilm - this is not correct, should be  Infections associated with bacterial biofilm formation

line 73 What S. epidermidis strain has been used? Specify the collection number or mention that clinical isolate has been used

line 86 It sound strange that biofilms were growing with shaking.

Line 88 It would be nice to have a scheme of the experiment 

line 91. Why 10 fold diluted broth has been use? This is additional nutrition depletion stress

Borders in Figs 2-5 should be removed, the the texture of bar also is recommended to remove

The english should be revized throughout the paper, and checked for typos

Round 2

Reviewer 1 Report

Comments and Suggestions for Authors

Dear authors,

Thank you for the revised version of the paper. You have addressed all my comments and suggestions for the manuscript. 

Author Response

From Reviewer 1: "Thank you for the revised version of the paper. You have addressed all my comments and suggestions for the manuscript."

We appreciate Reviewer 1's time and effort in helping us improve our manuscript.

Reviewer 2 Report

Comments and Suggestions for Authors

please add the method of distinguishing planktonic versus biofilm cells in the methods "Enumeration" section.

Author Response

From Reviewer 2: "please add the method of distinguishing planktonic versus biofilm cells in the methods "Enumeration" section."

We have revised Section 2.7 Enumeration to specify that immediately following treatment, the biofilms are separated from their surrounding planktonic bacteria by transferring them to a new 96-well plate with sterile media, then resuspending the biofilm for enumeration.  We also slightly modified Section 2.3 Thermal Shock to also indicate that separation.

Reviewer 4 Report

Comments and Suggestions for Authors

Authors made all required modifications, and answered almost all questions. 

Author Response

From Reviewer 4: "Authors made all required modifications, and answered almost all questions."

We appreciate Reviewer 4's time and effort in helping us improve our manuscript.